# Detecting IoT User Behavior and Sensitive Information in Encrypted IoT-App Traffic

**DOI:** 10.3390/s19214777

**Published:** 2019-11-03

**Authors:** Alanoud Subahi, George Theodorakopoulos

**Affiliations:** 1School of Computer Science and Informatics, Cardiff University, Cardiff CF10 3AT, UK; TheodorakopoulosG@cardiff.ac.UK; 2Faculty of Computing and Information Technology, King Abdulaziz University, Rabigh 25732, Saudi Arabia

**Keywords:** IoT, privacy, supervised machine learning, IoT privacy inspector

## Abstract

Many people use smart-home devices, also known as the Internet of Things (IoT), in their daily lives. Most IoT devices come with a companion mobile application that users need to install on their smartphone or tablet to control, configure, and interface with the IoT device. IoT devices send information about their users from their app directly to the IoT manufacturer’s cloud; we call this the ”app-to-cloud way”. In this research, we invent a tool called IoT-app privacy inspector that can automatically infer the following from the IoT network traffic: the packet that reveals user interaction type with the IoT device via its app (e.g., login), the packets that carry sensitive Personal Identifiable Information (PII), the content type of such sensitive information (e.g., user’s location). We use Random Forest classifier as a supervised machine learning algorithm to extract features from network traffic. To train and test the three different multi-class classifiers, we collect and label network traffic from different IoT devices via their apps. We obtain the following classification accuracy values for the three aforementioned types of information: 99.4%, 99.8%, and 99.8%. This tool can help IoT users take an active role in protecting their privacy.

## 1. Introduction

The Internet of Things (IoT) refers to the tens of billions of low-cost devices that communicate with each other and with remote servers on the Internet autonomously. It comprises everyday objects such as lights, cameras, motion sensors, door locks, thermostats, power switches and household appliances, facilitating our lives in almost every aspect of our day [1,2,3,4]. The most recent estimate is from GSMA Intelligence in June 2018, projecting “... over 25 billion IoT devices in 2025”, which is consistent with Gartner’s estimation in 2017 that by 2020 about 20 billion IoT devices will be connected to the Internet [5].

The diversity of IoT application domains is wide: smart cities, building and home automation, logistics and transportation, environmental monitoring, smart enterprise environments, and other smart wearable devices [6]. The recent rapid development of the IoT and its ability to offer a new platform for services and decision-making have made it one of the fastest growing technologies today. This new disruptive paradigm of a pervasive physically connected world will have a huge impact on social interactions, business, and industrial activities [7]. IoT wearable devices are predicted to reach a total of 126.1 million units in 2019 according to IDC, which will result in a five-year compound annual growth rate of 45.1% [8].

The proliferation of IoT, however, creates important security and privacy problem. IoT devices monitor, collect and store a huge amount of sensitive data and information about organizations, financial transactions, marketing insights, individuals, and product development [7]. For example, the popularity of wearable tech is one trend that is currently supporting much more extensive data capturing processes. Inventions such as the Apple iWatch, Google Glass, the Apple Health Kit, the Apple Home Kit, and Google Fit are constantly collecting information about the lives and habits of their users. It includes everything from financial data to information on medical conditions, physical fitness, shopping routines, music preferences, browsing behaviors, and much more [9]. However, when such sensitive personal data is released to third parties, the possibility of an unintentional or malicious privacy breach, such as detection of user activity, is very high [10].

Despite the importance of the privacy risk, the majority of IoT users do not understand what kind of information is being collected about them or their environment. In fact, a significant proportion of users are not fully aware that they are sharing their information in the first place [9]. The General Data Protection Regulation (GDPR) emphasizes that companies are required to protect the privacy of their EU customers by keeping Personally Identifiable Information (PII) secure. Personal data has been defined by the GDPR as follows: “Article 4(1): ‘personal data’ means any information relating to an identified or identifiable natural person (‘data subject’); an identifiable natural person is one who can be identified, directly or indirectly, in particular by reference to an identifier such as a name, an identification number, location data, an online identifier or to one or more factors specific to the physical, physiological, genetic, mental, economic, cultural or social identity of that natural person” [11,12].

There are two types of personal data: First, sensitive PII, which comprises information related to the user that is not for public use, or may violate the individual privacy and security by being made publicly available, e.g., log in details, telephone number, date of birth, full name, address. The other type of personal data is non-sensitive PII, which is information that can identify the user but will not affect his privacy or security, such as email address, first name, nickname, social media profile, website [13,14]. Privacy is not only about access authorization and encryption; rather, it also emphasizes on the type of transmitted information [15], and on how it will be used and shared by the legitimate recipient (e.g., IoT manufacturer) [16].

According to [17], there are three different methods of communication between the IoT device and its cloud: IoT device to IoT cloud (D-C); IoT mobile application to IoT device (A-D); IoT mobile application to IoT cloud (A-C). In fact, there are ample research efforts to uncover IoT security vulnerabilities and exploits [1,4,18,19,20,21]. However, researchers that address the privacy risks of IoT devices have focused on the traffic that goes directly from the IoT device to the IoT cloud (D-C) Path A in Figure 1. Nevertheless, a significant number of home-based IoT devices come with a companion mobile application. Each IoT manufacturer creates its own mobile application to control, configure, and interface with the device. Therefore, data from the IoT device can also reach the IoT cloud via the IoT app installed on the smartphone (Paths B and C in Figure 1).

To the best of our knowledge, no research studies this alternative path. Based on our analysis, the information that is being sent to the IoT cloud from the IoT app (path C) is much more sensitive than the information sent to the IoT cloud from the IoT device itself (path A) because this information not only reveals the type or the traffic rate of the IoT device, but also it could reveal users’ credentials, users’ location, or users’ current interaction with the IoT device via the app. The latter type of information is not evident from the traffic on path A.

In this work, we study the alternative data disclosure path C in depth. We invent an automated tool called IoT-app privacy inspector that analyzes collected encrypted traffic from IoT devices and classifies it using three supervised machine learning models. Each of them implements the Random Forest algorithm [22] and is used for a separate classification:
The first classifier classifies traffic by the type of app-device interaction (e.g., the user logs into the IoT app).The second classifies traffic according to whether it carries sensitive PII, non-sensitive PII, or non-PII.The third classifies PII traffic by the type of information it contains (e.g., user credentials or user location).

Once an attacker identifies a user’s interaction type e.g., log in to the IoT app, he can infer sensitive PII packets caused by this particular interaction; after that, he can infer the content type of such sensitive PII packet, e.g., log in credentials or geographical location. According to Wang et al. [23,24] 77.38% of users reuse one of their existing passwords. Also, Das et al. [25] estimate that 56% of users change their password at least once every 6 months because they tend to have the same passwords. This means that if an attacker manages to find the packet that contains the user’s password, he could mount an offline password attack to crack the password, which is impossible to detect and faster than an online attack. Therefore, he can gain access to every account the user has.

The contributions of our work are the following:
We show how passive packet-level analysis can be done to infer the behavior of the IoT device through the encrypted network traffic of its apps.We show how an attacker can infer the type of user interaction between the IoT app that controls the IoT device and the IoT manufacturer’s cloud (e.g., log into or log out from the IoT application).We show how an attacker can infer whether the IoT app sends sensitive PII to the IoT manufacturer cloud, as well as the kind of this sensitive PII, caused by user interactions.

The rest of the paper is organized as follows: Section 2 highlights recent research in IoT traffic classification as well as in IoT privacy. In Section 3, we discuss how an attacker could attack and collect smart-home traffic in our attacker model, followed by a detailed description of the method we use to establish the ground truth in Section 4. In Section 5, we present our attack design and implementation, while in Section 6 we develop our inspector tool with three multi-class classification methods, each one used to infer a different goal; we also evaluate our tool. We present the limitations of our research and discuss future work in Section 7, followed by a summary and conclusion in Section 8.

## 2. Literature Review

Most of the security and privacy research regarding IoT devices has focused on security issues [26,27]. Other security research monitors IoT traffic to detect intrusion attempts [28] or has discovered various IoT vulnerabilities [6,29,30,31,32]. However, the contribution of this paper is to emphasize on the risks and vulnerabilities associated with the type of personal information being collected from the IoT app and sent to the IoT cloud and whether such information could reveal sensitive user activity. In this section, we first examine the research that classifies IoT traffic. Second, we examine the most relevant research to ours, which is privacy research that monitors IoT network traffic to infer sensitive information contained in the traffic.

### 2.1. IoT Traffic Classification

Even though there is a huge body of work characterizing general Internet traffic, research focusing on characterizing IoT traffic (also called machine-to-machine—M2M—traffic) is still in its infancy. One of the first huge-scale studies to investigate the nature of M2M traffic has been done by Shafiq et al. [33]. They want to understand whether IoT traffic imposes new challenges for cellular networks in terms of their design and management. [34] has suggested that vast quantities of IoT device information reflecting common behavior and a sole IoT device’s communication behavior can be determined through a Coupled Markov Modulated Poisson Processes model.

The profiling, evaluation, and categorization of smart context IoT devices was undertaken in [35], through analyzing 21 distinct IoT devices. Three weeks of data from traffic traces were obtained during the research, which was then put in the public domain. Subsequently, the protocols, signaling, activity trends, and other features of the traffic were statistically assessed. Ultimately, a classification method was devised with a greater than 95% precision rate for determining specific IoT devices, in addition to being able to ascertain whether the device was IoT-enabled or otherwise.

A logical IoT device classification model was developed in [36]. However, their model was limited to only classify the IoT devices into two categories, namely high vs. low energy consumption, so as the authors state, it is still at a primary stage.

Furthermore, the precise categorization of non-IoT and IoT devices based on assessing traffic, which also enabled the specific model and company to be identified, was undertaken using machine learning methods in [37]. Two smartphones, two computers and 10 IoT devices situated locally were used to obtain traffic information with a 99.281% precision rate in the classification.

All previous research deals with IoT traffic classification with the aim of (1) classifying the IoT traffic from the non-IoT traffic, (2) classifying the IoT traffic to determine specific IoT device, (3) classifying the IoT traffic into low energy or high energy in order to understand the current behavior of the IoT device. In contrast, in our research we classify IoT traffic aiming to accurately infer (1) if the packet reveals the interaction type between the IoT device and its corresponding IoT app, (2) if the packet reveals sensitive Personal Identifiable Information (PII) about their user, and (3) if the packet reveals the content type of such sensitive information.

### 2.2. IoT Privacy Concerns

Motivated by privacy issues, Apthorpe, Reisman, and Feamster [38] showed how a passive network observer, e.g., an Internet Service Provider (ISP), can analyze traffic data to infer sensitive information about consumers as well as the type of connected IoT device even when the traffic is encrypted. They examine four commercially available IoT smart-home devices and find that an IoT device’s particular activity and its type can be revealed through network traffic rates by anybody passively monitoring the traffic rate pattern. For example,
An Amazon Echo’s traffic [39] can indicate when the intelligent personal assistant is being engaged with by a user.Motion detection by a camera, as well as a user is observing its live images, can all be determined from a Nest Cam [40] Indoor CCTV’s traffic levels.Whether a Belkin WeMo switch device [41] is on or off in a smart house can be inferred from its traffic.The sleep pattern of a user can be understood from a Sense sleep [42] device’s traffic levels, by someone monitoring such traffic.

Our work is similar to the above in that we also study and analyze the encrypted traffic of the IoT devices. However, in that research, the focus was on (1) studying the traffic that goes directly from the IoT device to the IoT cloud (D-C), i.e., Path A in Figure 1, (2) studying only the *traffic rate* pattern to infer the type and the activity of the IoT device. In contrast, our focus is on the traffic that goes directly from the IoT app that controls the IoT device to the IoT cloud (A-C), i.e., path C in Figure 1. Also, we do a more in-depth analysis by examining the *size and sequence of the packets*, and because of this we are able to infer the user interaction with the IoT device (e.g., login to the IoT device), the user sensitive data, and finally the type of such sensitive data (e.g., password).

Siby et al. [43] developed a system called IoTScanner to analyze the IoT environment. The low energy frequencies of Bluetooth, Zigbee and, Wi-Fi used to transfer traffic can be monitored through this system. Furthermore, IoTScanner gives an overview of the active and operating IoT devices within an environment, in addition to their intercommunication. As a result, they find that it is possible to violate user privacy by classifying active Wi-Fi IoT devices, via the ratio of the send and receive traffic. In our research, we prove that user privacy can be violated by monitoring the IoT traffic to determine the user behavior with the IoT device via its app as well as to determine the sensitive information about the user.

Traffic levels were used to determine IoT behavior in a further trial [44], where it was determined that encryption processes would still enable packet headers and smart-home traffic levels to be used by a passive network attacker to determine local activities.

As a result of previous research, the correlation between traffic patterns and sensitive activities motivated us to apply machine learning to infer the three different classifications mentioned earlier.

## 3. Attacker Model

We consider a passive network observer who accesses smart-home traffic. We assume that our adversary can collect the transport-layer traffic of a smart home. Also, we assume that packet contents are encrypted using TLS. This adversary can be the ISP, which can collect and store traffic regularly, or, in general, it can be any adversary who knows the SSID and the WPA2 password of the smart-home router. Finally, the adversary can get a database of labeled traffic from smart-home devices for training machine learning algorithms.

The adversary’s goals are the following:
(A)Infer the user’s interactions with IoT devices in a smart home (e.g., logging into the smart-plug app),(B)Determine whether the transmitted data carries sensitive personal identifiable information (PII), non-sensitive PII, or non-PII about the user,(C)Determine the type of sensitive PII (e.g., password for the IoT device app) or non-sensitive PII (e.g., user email) that is being transmitted.

As the traffic is TLS-encrypted, the adversary must rely on traffic rate, packet size, and packet sequence information to make any inference; he cannot read the packet contents. This inference is especially worrisome as it is completely passive, and so it would require no change to existing data collection procedures.

## 4. Methodology

Before we explain our laboratory smart-home environment, we recall the two different ways that IoT devices use to communicate with their manufacturer’s cloud:
Device-to-cloud: as Figure 1 in the Introduction illustrates, path A represents the direct data transfer from the IoT device to the IoT cloud. Most research, to the best of our knowledge, focuses only on this path to study the contents, patterns, and metadata of IoT network traffic that reveals sensitive information about user activity [34,38,44]. However, this type of information does not violate user privacy as the second way (A-C) does.App-to-cloud: all IoT devices are controlled and configured via their mobile apps [35], no two IoT devices from two different manufacturers are sharing the same app. For example, a TP-link smart plug is controlled by a mobile application called KASA, while a WeMo smart plug is controlled by a different mobile application called Wemo. These mobile apps are recommended by the IoT device manufacturers, and installed on the smart phone or tablet to control the IoT device [35]. In a typical scenario, as in Figure 1 paths B and C, when a user wants to switch on/off a smart plug, he first needs to log in to the IoT app and then press the switch on/off button. In this case, a command is sent to the smart plug via its app to switch on/off. In parallel, traffic with sensitive personal information is sent to the smart-plug cloud from the IoT app to inform that the user has logged in to the app and switch on/off the smart-plug.

In this research, we focus on collecting and analyzing the data transferred from the IoT app to the IoT cloud. Please note that many IoT devices use TLS/SSL when communicating with cloud servers, so the traffic we collect is encrypted. Given the increasing focus on security in the IoT community, we expect that encrypted communications will become standard for smart-home devices.

### 4.1. Overview of the IoT-App Privacy Inspector Tool

The tool takes as input encrypted traffic collected from different IoT devices. The first classifier classifies the packets according to whether they contain sensitive PII or non-sensitive PII, or none. The second classifier will classify the content type (user credential, location, username) of such sensitive or non-sensitive PII packets. Finally, the third classifier classifies the packets based on the user interaction type with the IoT device (login, logout, delete a device, change password). Figure 2 gives an overview of the proposed tool.

### 4.2. IoT Smart-Home Testbed

We set up our smart-home testbed with four well-known and commercially available IoT devices as a representative example of a smart-home. The devices included in this testbed are TP-link smart plug (https://www.tp-link.com/uk/home-networking/smart-plug/hs100/); TP-link smart camera (https://www.tp-link.com/uk/support/download/nc200/); Belkin NetCam (https://www.belkin.com/uk/p/P-F7D7601/); Lifx smart bulb (https://uk.lifx.com/products/lifx). An Android smartphone was also connected to the network. We install the recommended apps on the smartphone to control the functions of each IoT device in our testbed; see Table 1 for more details. Additionally, a laptop (running Kali Linux) was also connected to the network to perform two tasks: (1) monitor and continually collect the network traffic between the IoT device and the smartphone app, and also between the smartphone app and the cloud, and (2) perform a Man in the Middle attack (MITM) as we explain in Section 5. Figure 3 displays the architecture of the smart-home testbed.

## 5. Attack Design and Implementation

The steps below give a high-level description of our implementation, which is what the attacker would do:
Select IoT devices whose traffic should be classified by the tool.Establish ground truth about user interactions with the IoT devices by doing the following steps:
(a)Collect IoT traffic while performing various interactions with each device to generate traffic.(b)Analyze the IoT traffic in order to identify the interaction type, the packets containing sensitive PII, non-sensitive PII, and non-PII, and within the PII traffic (both sensitive and non-sensitive) identify the content type (e.g., user credentials or username).(c)Annotate the traffic by labeling each packet with the interaction type that created it.Use the labeled traffic as training data for a classifier to infer the three goals stated above (Section 3) from unlabeled/unseen traffic. This point will be explained in detail in Section 6.

Now we explain step 2 in the following subsections in more detail.

### 5.1. Establish Ground Truth

#### 5.1.1. IoT Traffic Collection

We conduct our experiments to establish ground truth from November 2018 until April 2019. An overview of our experiments can be seen in Figure 4. We use Wireshark [45] to passively capture and collect the traffic data of the IoT devices and their relevant IoT apps: First, we determine the IP address of each IoT device within the smart-home network; then, we identify the IP address of the smartphone that has the installed IoT apps. The second and third steps are performed in parallel: In the second step, we intercept and therefore collect the traffic by conducting a MITM attack (with ARP spoofing) [46] between the smartphone and the IoT cloud. This attack allows us to record all network traffic between the IoT cloud and the IoT app in both directions. Figure 5 and Figure 6 illustrate the redirection that ARP spoofing causes in the traffic between the IoT app and the IoT cloud. Before ARP spoofing, the traffic goes via the router, Figure 5; after the ARP spoofing, the traffic goes via the attacker device (in this case via our Kali laptop), then Kali sends it to the router as Figure 6 shown.

While the MITM attacks are active, we interact with each IoT device mentioned in Table 1 separately. We perform four different interactions because they are common among IoT apps. These actions are the following:
Login to the IoT application, permitting the user to control the IoT device functions;Alter settings including changing the password, permitting the user to change the IoT device settings or the password;Delete the IoT device, allowing the user to cease use of the IoT device by deleting it from the application, and consequently deleting it from the IoT cloud/server database;Logout from the IoT application, which sends the user’s access or control of the IoT device functions.

This second step collects encrypted TLS traffic that we need to decrypt to establish the ground truth about the packets that the IoT app sends to the IoT manufacturer’s cloud. We do this decryption in the third step, while we are collecting the traffic. First, we used Burp Suite tool [47] on our Kali laptop. In Burp Suite, we set up the proxy server port to 8080 to listen to the network traffic of the smart phone and the IoT device. Second, we configure the Wi-Fi setting of the smartphone to use the same proxy server port. Finally, we install the Burp Suite certificate onto the smart phone User Trust Store.

It should be noted that these steps only work if the IoT app does not employ certificate pinning [48]. In our case, KASA, TpCamera, and NetCam do not employ certificate pinning, but Lifx does. One of the solutions to solve the certificate pinning problem is to reverse engineer the IoT app. Then, install the fake certificate from Burp Suite. Finally, recompile the new version of the IoT app and re-install it on the smart phone.

#### 5.1.2. Activity Inference from Collected Traffic and Identification of Packets Comprising User Interaction, Sensitive PII, and the Content Type of the Sensitive PII

In this section, we present our observations from a passive packet-level analysis of collected traffic from the IoT devices installed on the smart-home testbed. As we explain in the previous section, for each of the four interactions with each of the four IoT devices, we collect one encrypted pcap file from Wireshark and one corresponding decrypted burb file from Burp Suite. To analyze and therefore identify the type of user interaction with the IoT app, the packet sensitivity level, and the packets that contain personal information along with the type of personal information that they contain. We analyze each pair of burb file and pcap file for each interaction with each IoT device separately.

##### Analyzing the Burp Suite Files

We establish the ground truth about the user’s interactions with the IoT devices by analyzing the decrypted traffic we obtain from Burp Suite file of each IoT app. In particular, we correlate the actions that the user invokes on the IoT device with the packet sizes and sequences that result from these actions.

We find that each IoT app communicates with several domain names associated with the IoT device manufacturer. Interestingly, we realize that each domain name is responsible for certain types of interaction. For example, the KASA and TpCam apps from TP-link communicate with two different domain names, while the NetCam app from Belkin communicates with three different domain names, and finally the LIFX app from Lifx communicates with five different domain names.

Figure 7 illustrates an example of the two domain names that KASA app communicates with, which are api.tplinkra.com and eu-wap.tplinkcloud.com, both owned by TP-link. Each domain name is responsible for a particular set of methods. See Appendix A for the rest of IoT-app domain names. For example, each time the user logs in to KASA app, the methods listed in Table 2 are executed, always in the same sequence. Each method always generates a request packet from KASA app to the domain name responsible for this method. It is followed by a response packet from that domain name to KASA app, with the indicated packet sizes and sequences. In Table 3, we observe the sequence and the packet sizes of the methods that are executed when the user logs out from KASA app, and we see that they are different from Table 2. We observe similar differences for the other actions of this and the other IoT devices; see Appendix B. Because these sizes and sequences are unique to each action, an attacker can use them to identify the invoked actions. Also, because each packet in a sequence always contains the same type of information, the attacker can detect the packets that contain sensitive information.

Based on these findings, we conclude that we can rely on the packet sizes and sequences to infer whether the user interaction with the IoT app is login, logout, and so forth. Furthermore, we manage to identify the length of every packet that sends to or receives from the IoT cloud any personally identifiable information (e.g., user location, username and password). For example, we can confirm that any packet sent by the KASA app with a packet size of 520 bytes and a received size of 873 bytes from the TP-link domain name eu-wap.tplinkcloud.com, is the passthrough method. This method is always triggered when the user logs in to the KASA application, and it carries information regarding the user’s geographical location, see Figure 8; similarly for the remaining methods.

In some cases, we notice that the packet sizes do vary across executions of a method. This variation is small and thus does not affect our classification negatively, but it can reveal additional information. For example, the size of the request packet for the login method, see Table 2, is always 542 bytes plus the length of the user’s password. This means that the password length is only 6 bytes in this example. From a security perspective, this is an important finding because the attacker can determine the password length, and therefore determine whether a brute force attack is feasible to obtain the password. Please note that this attack can be done offline, so any measures on the IoT cloud side to block repeated failed password submissions would not help.

##### Analyzing the Wireshark File

We now aim to match the encrypted packets from the Wireshark file to the equivalent decrypted packets from the Burp Suite file. We can then label each packet of the encrypted traffic and use this labeled traffic to train our machine learning classifier. The most straightforward way to do this match would be to match encrypted packets to decrypted packets of the same size. However, the sizes of encrypted and decrypted packets are not similar, so we design a new method to find this match. We apply our method to all actions of the IoT apps. We describe this method in the steps below, in which we aim to match encrypted-decrypted packets for the logout action in KASA app as an example.
(a)First, we filter the pcap file to keep only the packets whose source IP address belongs to the smartphone that has the IoT app, and whose destination IP address belongs to one of the two IoT domains of KASA app, see Table 3.(b)Then, in the pcap file, we look for a sequence of encrypted packets whose source and destination IP addresses match the corresponding sequence in the methods of the logout action in the decrypted packets from burp file. For example, the user logout action from KASA app triggers five methods. Therefore, in the pcap file, we expect to find the same five5 methods in the same order. In Table 3, the first method in the logout action is logout method, which communicates with the eu-wap.tplinkcloud.com server, followed by the second method helloIotCloud, which in turn communicates with the pi.tplinkra.com server and so on for the rest of the methods. Therefore, we should find in the pcap file the same domain names in the same order. As we mentioned earlier, each domain name is responsible for specific methods. By finding the same sequence of the domain names, we can prove that we have found the correct expected method.(c)After identifying the correct method, we now want to match the actual packets. We compare the request and the response packet size of the logout methods from the pcap file with the response and the request packet size of the equivalent logout methods from the burp file. We find that encryption always adds a constant number of bytes to the plain packet size:
The size of the encrypted packet for the logout method request is equal to the decrypted packet size plus 148 bytes (decrypted: 521 bytes; encrypted: 669 bytes).Similarly, for the response traffic, the encrypted packet size is equal to the decrypted packet size plus 95 bytes (decrypted: 178 bytes; encrypted: 273 bytes).We observe the same constants (148 bytes for request packets and 95 bytes for the response packets) for all packets of the KASA app. We link this constant to the type of cipher suite that KASA app use, which is TLS-ECDHE-RSA-WITH-AES-128-GCM-SHA256. Other apps also exhibit the same behavior, only with different additive constants for their request and response packet sizes, because they have different cipher suite. For example, netcam app uses TLS-RSA-WITH-AES-128-CBC-SHA cipher suite.(d)Finally, as a visual verification step that we match the correct packets, we create a plot per decrypted action and a corresponding plot per encrypted action. By comparing the two plots, we find that they are equivalent. Figure 9 illustrates the logout action and the method sizes and sequences from the burp file from KASA app. After applying our method, we find the same methods with the same order in the pcap file, as you can see in Figure 10. Note the packet sizes are for encrypted packets. The plots for the rest of the actions can be found in Appendix C.

##### Feature Selection and Data Labeling

During this stage, we compose all packets that are transmitted between the same pair (IP-src, IP-dst) to a group of sessions. Next, we select the most important features that help us manually label all the encrypted session according to the following categories:
the user interaction with the IoT device that the packet is part of;whether they contain sensitive information;the content type of the packets that contain sensitive information.

These features are the following:
IP-src: refers to the IP address of the smartphone running the IoT app;IP-dst: refers to the IP address of the IoT-app domain.Comm-type: refer to which domain name the IoT app communicates with (e.g., KASA app communicate with two domains, so if the IP-src belongs to the smartphone and the IP-dst belongs to the second domain name, then the comm-type set to 1.2);Req-len: refers to the length of the sending packet (from the IP-src to the IP-dst);Resp-len: refers to the length of the receiving packet (from the IP-dst to the IP-src).

We label the sessions in three different ways, thus creating three different datasets. Each one is used to train and test one classifier—see Figure 11. For the first dataset, named IoT-interactionType, we label the packets according to the interaction type between the user and the IoT app with either “Login”, “Logout”, “Change Password”, “Delete”, or “None”. For the second dataset, named IoT-PII, we label the packets according to their sensitivity level with either “Sensitive PII”, “Non-sensitive PII”, or “None”. For the third dataset called IoT-user-PIItype, we label the sensitive packets (sensitive PII or non-sensitive PII) according to their content type with either “User credentials”, “User location”, “username”, or “None”.

Once an adversary creates or obtains such labeled traffic for the IoT devices of his choice, he can create a classifier to identify packet streams pertaining to a specific IoT device. Then, he can infer a specific user interaction in unlabeled traffic. Therefore, he will be able to infer the packets that carry sensitive information and the content type of this sensitive information. In the next section, we describe the design of the classifiers.

## 6. Machine Learning-Based Classification

We treat the tasks of identifying user interaction type, packet sensitivity level, and sensitive data type as a multi-class classification problem. Accordingly, six classifiers were selected based on their ability to support multi-class classification.

To evaluate the performance of the selected algorithms and hence choose the best classifier for our problem, we apply several measures. The most common measures are precision, recall, F-mean, and accuracy. As an example, the first multi-class classification problem is evaluated relative to the training dataset, producing the following four outputs:
**True positive (TP)**—packets are predicted as a sensitive PII, when they are truly sensitive PII.**True negative (TN)**—packets are predicted as a None when they are truly None.**False positive (FP)**—packets are predicted as sensitive PII, when they are truly None.**False negative (FP)**—packets are predicted as None when they are truly sensitive PII.

Precision (P) measures the ratio of the packets that were correctly labeled as sensitive PII to the total packets that are truly sensitive PII *[Precision = TP/(TP + FP)]*. Recall (R) measures the ratio of the packets that were correctly labeled as sensitive PII to the total of all packets *[Recall = TP/(TP + FN)]*. F-measure (F) takes both false positives and false negatives into account by calculates precision and recall. Then, it provides a single weighted metric to evaluate the overall classification performance *[F1 Score = 2 × (Recall × Precision)/(Recall + Precision)]*. Accuracy measures the ratio of the packets that were correctly predicted to the total packets number of the packets *[Accuracy = (TP + TN)/(TP + FP + FN + TN)]*. However, using accuracy to measure the performance of a classifier is a problem. This is because if the classifier always infers a particular class, it will achieve high accuracy, which makes it useless when it comes to building such a classifier.

The goal is to maximize all measures, which range from 0 to 1, to achieve better classification performance. Table 4 illustrates the overall results based on previous measurements. As we can see, the Random forest exhibits the best performance across all six classifiers. Therefore, we develop our classification tool based on the Random Forest classifiers. To support our choice, a recent survey on ML methods for security [49] discusses the advantages of using Random Forest. Their study is related to our research as it combines decision-tree induction with ensemble learning; these advantages are:
Very fast when classifying input dataResilient to over-fitting.It takes a few input parameters.The variance decreases as per the increment of tree numbers, excluding any biased results.

### 6.1. Multi-Class Classifier Training

To perform our classification experiments, we randomly split each dataset described in Section 5.1.1 into 80% for training, and the remaining 20% for testing. Notice that each classifier applies to one dataset; see Figure 11. Each classifier is responsible for inferring the possible label of one category. As we can see in Table 4, the Random Forest classifier achieves the best performance resulting in 99.8%, 99.8%, and 99.8% in the first and the second classifier, while it achieves 99.4%, 99.4%, and 99.4% in the third classifier for the measurements of precision, recall, and F-mean score, respectively. Additionally, the classification time is 0.35 s, for each classifier.

To validate that the classifier does not overfit, we perform several experiments:
**10-fold cross validation experiments**To determine the optimal hyperparameters of the Random Forest algorithm [22,50], we try many different combinations using GridSearch algorithm optimization. Based on the results, we set our hyperparameters as follows: the number of n-estimator is 10, min-samples-leaf is 3, bootstrap is “False”, min-samples-split is 8, criterion is “entropy”, max-features is “auto”, and the max depth is 90.**Confusion matrix experiments**To get a better understanding of the performance of the classifier across the experiments, the confusion matrices of the three classifiers in Table 5, Table 6 and Table 7 consecutively show the predicted classes for individual packets compare against the actual ones. Every confusion matrix is a synopsis of inferring the outcome of one multi-classification problem, which demonstrates the process in which our classification model is confused upon making an inference. Then correct and incorrect inference numbers are summarized through count values and decoded to each class. The individual confusion matrix gives us an in-depth look into errors being made by a classifier and mainly focuses on the sort of errors being made. For example in Table 5, the confusion matrix which is related to inferring the user interaction, shows that the actual number of the Delete interaction sessions is 284. However, the classifier correctly infers 281 sessions as a Delete interaction, while it infers incorrectly two packets as Logout interaction and one packet as No-action. These results confirm the high accuracy and reliability of our classifiers.**Compare the accuracy of the training dataset with the accuracy of the testing dataset**The training accuracy is the accuracy of the classifier on the training dataset, while the testing accuracy is the accuracy of the classifier on the testing dataset. If the accuracy of the training data is almost similar to the accuracy of the testing dataset, then there is no over-fitting issue; otherwise, we have an over-fitting issue. Table 8 shows that the accuracy of the training dataset and the accuracy of the testing dataset are very similar in all the three classifiers.

As a result of the previous experiments, we conclude that the IoT-app privacy inspector tool does not fall into the over-fitting problem.

### 6.2. Results and Discussion

An overview of the steps of the IoT-app privacy inspector tool is outlined in Figure 12. At first, the tool receives collected unseen IoT traffic in a pcap file format. Next, it extracts the relevant features from the pcap file as mentioned earlier (Section 5.1.1). Three different classifiers will be applied to this dataset. Each one is used for different inferences (Figure 11).

#### Unseen Validation Datasets

To evaluate the performance of our tool, we apply the trained classifiers to unseen datasets. We collect such datasets in Section 5.1.1 to validate the classifiers. Notice that we did not include the validation dataset in the original dataset used to train our classifiers. Accordingly, we conduct two types of evaluations to evaluate the accuracy and reliability of the IoT-app privacy inspector tool.

##### Classification Accuracy for Each IoT-App Interaction Separately

In the first evaluation experiment, we test the tool on each IoT device individually (one IoT device each time). For each IoT device, we apply the tool four times, on a collected dataset for each interaction Login, Logout, Delete, and Change Password. Thus, we apply the tool 16 times in total.

The results show that in every experiment the tool infers the correct class. We summarize and group the results from the 16 experiments according to each IoT app in Table 9. Each row represents one user interaction and the output of the IoT-app inspector tool (the three classifiers). For example, in the first row, the IoT-app inspector tool accurately infers that when the user logs into to KASA app, only sensitive PII packets are sent to the IoT cloud. The type of these sensitive packets is user credentials and user location.

In Table 10, we compare the results of all user interactions with all IoT devices. Our findings show that most interactions are similar in terms of sending sensitive PII or non-sensitive PII packets to their IoT cloud. However, we highlight three important things. First, the change-password interaction and the login interaction send both sensitive PII and non-sensitive PII packets to the IoT cloud from Lifx app. This means that Lifx app excessively sends sensitive PII packets about their user to the Lifx cloud through these two interactions. Second, logout interaction from netcam app does not send any type of sensitive packets to its IoT cloud, which makes it the safest interaction among the others. Finally, the delete interaction and the logout interaction of KASA, TpCam, and Lifx send only non-sensitive PII packets to its IoT cloud. Hence, these two interactions are seen to be the interactions that least send sensitive PII packets about the user to the IoT cloud.

##### Classification Accuracy with Mixed IoT Interactions in the Same File

In the second evaluation experiment, we test the tool four times on each IoT device individually (one IoT device each time). For each IoT device, we apply the tool on mixed user interactions between the IoT app and its IoT device to validate the classification accuracy by inferring the previously mentioned aims. The results presented in Table 11 demonstrate very high classification accuracy of our three classifiers:
the average accuracy (number of correctly inferred user interactions divided by the total number of interactions) is 99.4% with F1 score 0.994;the average accuracy (number of packets for which the level of sensitivity is correctly inferred divided by the total number of packets) is 99.8% with F1 score 0.998;the average accuracy (number of packets for which the content of the sessions correctly inferred divided by the total number of packets) is 99.8% with F1 score 0.998.

As a result of the previous experiments, we prove the validity and reliability of such a tool. We achieve high accuracy for inferring the correct type of sensitive information, as well as for inferring the user interaction type that occurs between the IoT device and the user.

## 7. Limitations and Future Work

Our method is subject to the following limitations:
Only devices communicating via TCP/IP were studied. Protocols such as ZigBee and Bluetooth were not included even though they are employed by some IoT devices.We collect benign IoT traffic, i.e., we do not compromise the IoT device nor use it in an unusual manner. Our conclusions therefore apply only when capturing normal behavior patterns of diverse IoT device types.

In future work, we plan to diversify the devices used in our lab and extend the device type identification and IoT-app privacy inspector method to additional communication protocols. We are also looking at ways to automate part of the labeling process, to make it easier to train classifiers for new IoT devices. Finally, we plan to make this tool available as an open-source mobile application tool that passively monitors and collects the traffic of any installed IoT app.

## 8. Conclusions

In this research, we start with the observation that there are two different ways of sending information about the IoT user to the IoT cloud: Device-to-cloud and App-to-cloud. To the best of our knowledge, no research has been done on the second way i.e., App-to-cloud. We show that any adversary who can observe and collect smart-home traffic can reveal sensitive information about the IoT user through the packet sizes and the packet sequences. For example, the adversary can infer, in real time, that a specific interaction (e.g., login to the IoT app) is occurring between the user and a smart plug via its related IoT app. In addition, the adversary can infer which packets carry sensitive information about their user, as well as the type of this information (e.g., user location or user credential).

We build a multi-class classification tool called IoT-app privacy inspector using supervised machine learning to raise the awareness of the IoT users about specific interactions that cause a violation of their privacy. For training data, we label the encrypted TLS transport-layer traffic that is being sent to the IoT cloud from the IoT app. We want the tool to be able to
classify the interaction of the user with every IoT-app (e.g., log in to/log out of the IoT-app);classify the packets generated by the user interaction according to their sensitivity level (e.g., sensitive PII, non-sensitive PII, non-PII)classify the content of the sensitive PII (into e.g., user credentials, user location) and the content of the non-sensitive PII (into e.g., user email, username).

We leverage the observation that the traffic generated by IoT apps follows a limited set of patterns, which allows us to perform the three classifications above. After training, this tool can be continuously applied to classify newly collected (unlabeled) IoT device traffic data.

Our tool aims to help IoT users by notifying them of any interactions that send excessive personal data to the IoT cloud e.g., when they login to the IoT app. The tool can accurately detect the TLS traffic that originates from any IoT app that controls the IoT device. Then it infers the user interaction type with the IoT app, infers whether there is any sensitive PII packet being sent to the IoT cloud and infers the type of the sensitive PII packet (e.g., user credentials). The results show that 99.4% of the user interactions with the IoT app are correctly detected, while 99.8% of the packets the carry sensitive PII caused by this interaction are correctly detected. Finally, 99.8% of the content type of this sensitive PII packets are correctly detected. The high accuracy results achieved by our tool prove the reliability of such a tool. Finally, we point out a security problem: It is possible for an attacker to identify the packet that contains the user’s password, and thus to launch an offline password cracking attack.

## Figures and Tables

**Figure 1 sensors-19-04777-f001:**
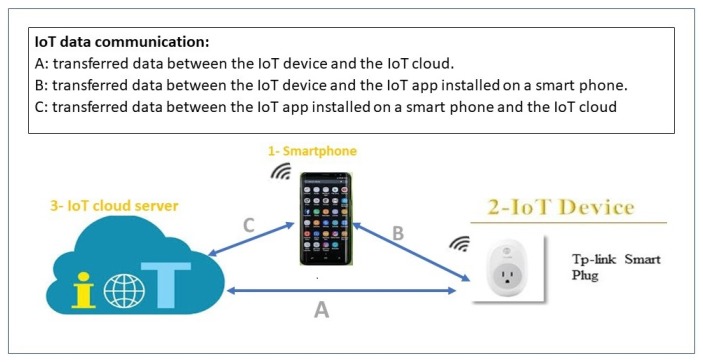
Methods of IoT communication with its cloud to transfer data.

**Figure 2 sensors-19-04777-f002:**
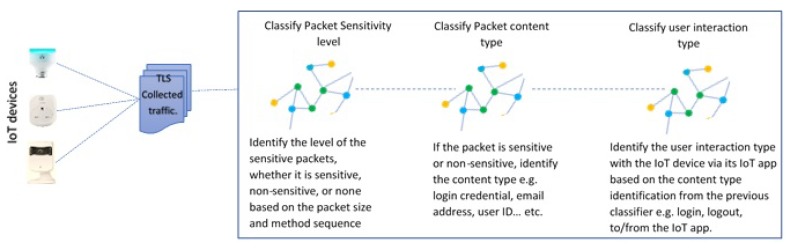
IoT-app privacy inspector tool overview.

**Figure 3 sensors-19-04777-f003:**
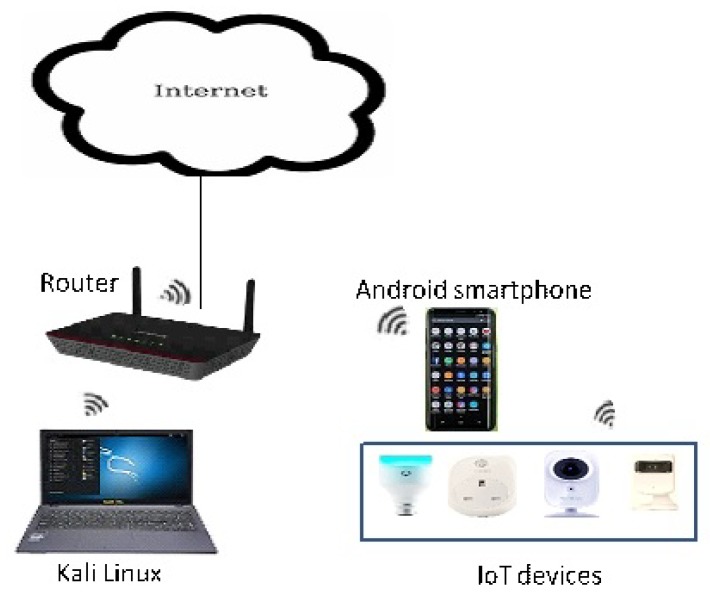
IoT smart-home testbed network architecture.

**Figure 4 sensors-19-04777-f004:**
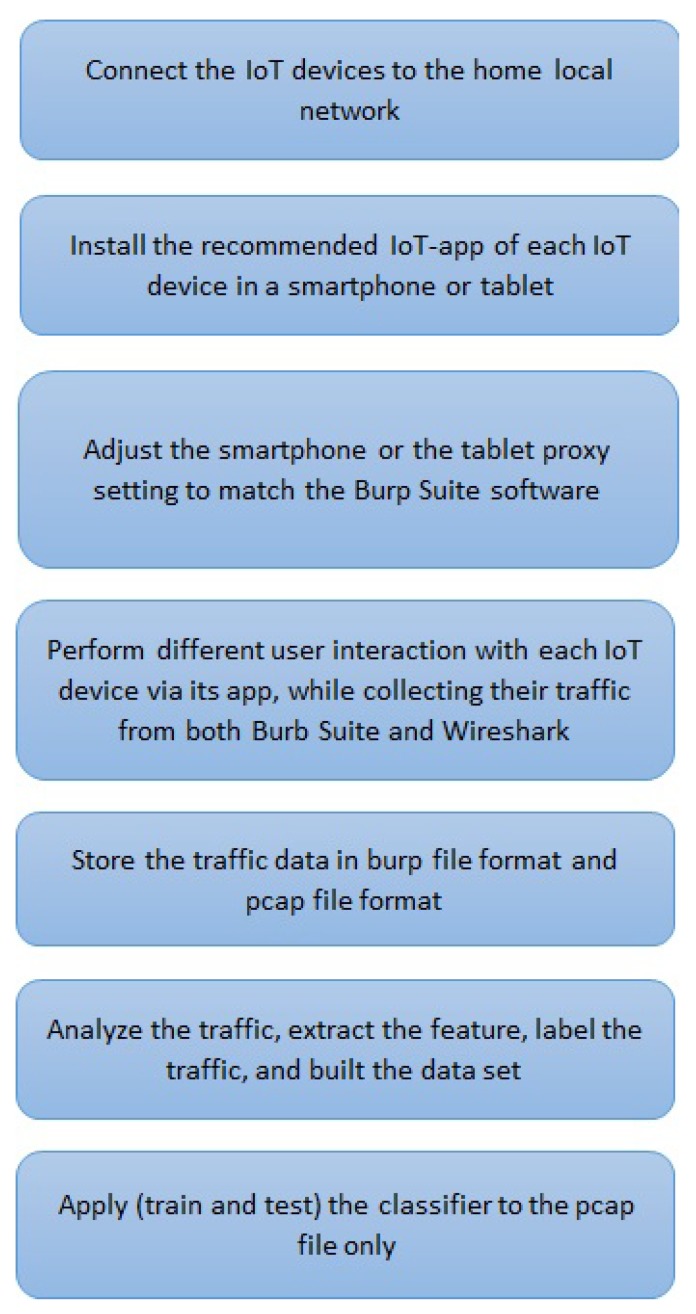
Overview of the steps used to collect the encrypted TLS traffic and the encrypted one of the IoT device to establish the ground truth of the IoT-app privacy inspector tool.

**Figure 5 sensors-19-04777-f005:**
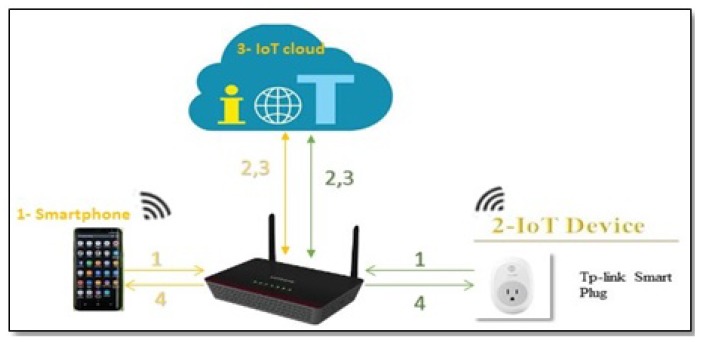
Normal traffic: All traffic goes through the router.

**Figure 6 sensors-19-04777-f006:**
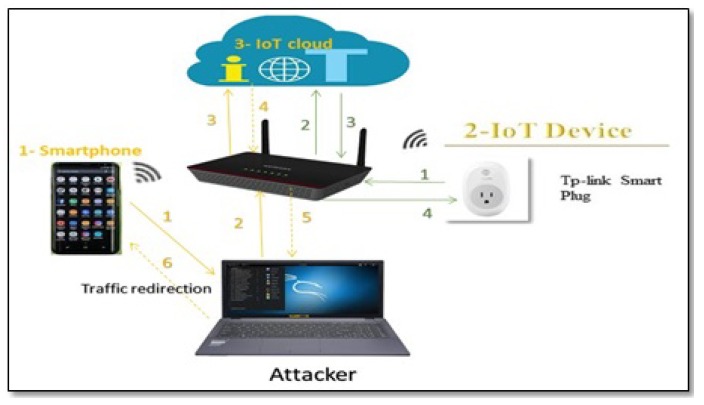
Spoofed traffic: IoT-app traffic is redirected through the attacker.

**Figure 7 sensors-19-04777-f007:**
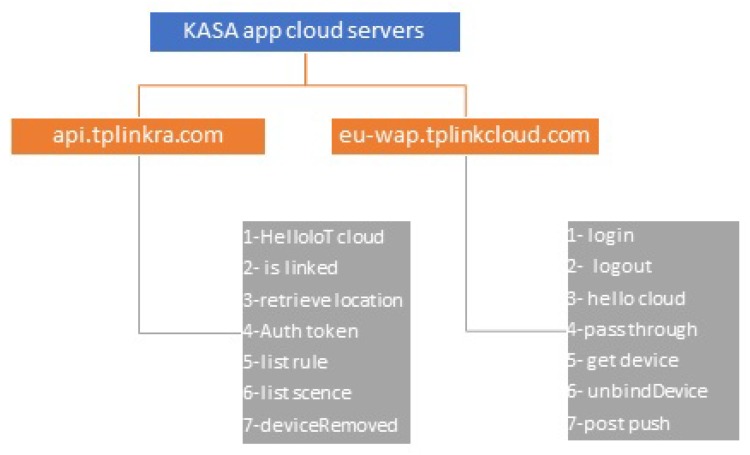
TP-link smart-plug domain names that KASA app communicates with. Each domain responsible for specific methods.

**Figure 8 sensors-19-04777-f008:**
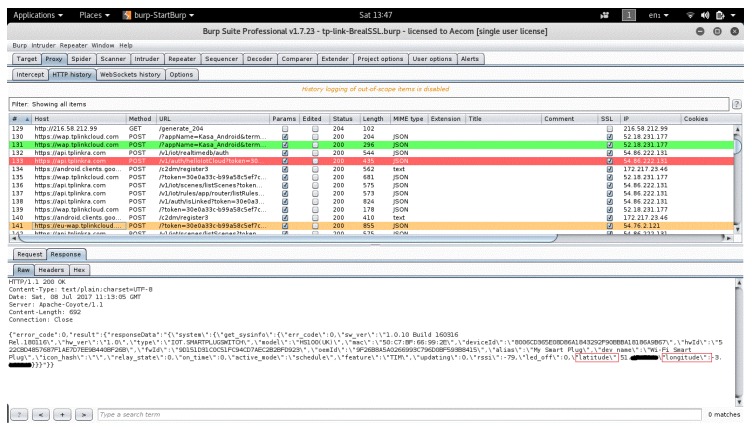
Screen shot from Burp Suite showing user’s exact location (latitude and longitude).

**Figure 9 sensors-19-04777-f009:**
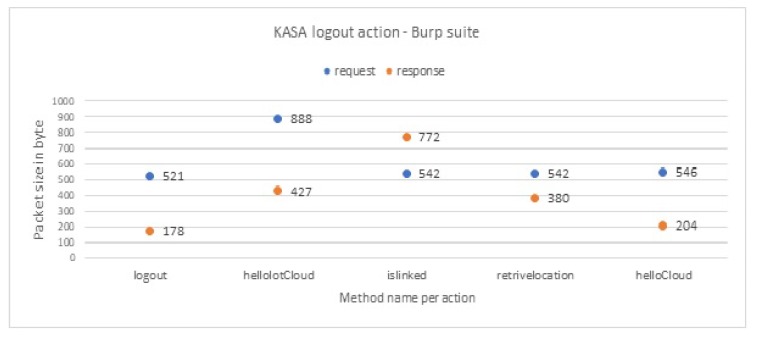
User logout interaction from KASA in decrypted format.

**Figure 10 sensors-19-04777-f010:**
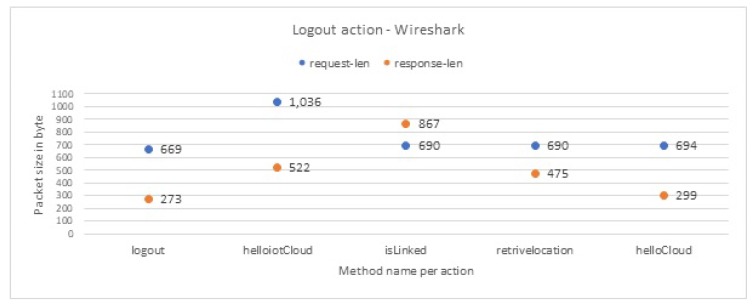
Equivalent user logout interaction from KASA in encrypted format.

**Figure 11 sensors-19-04777-f011:**
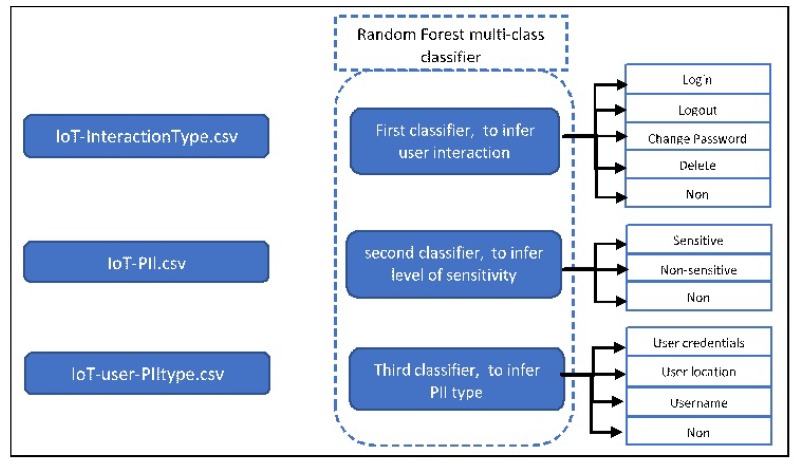
Overview architecture of the multi-class classifier.

**Figure 12 sensors-19-04777-f012:**
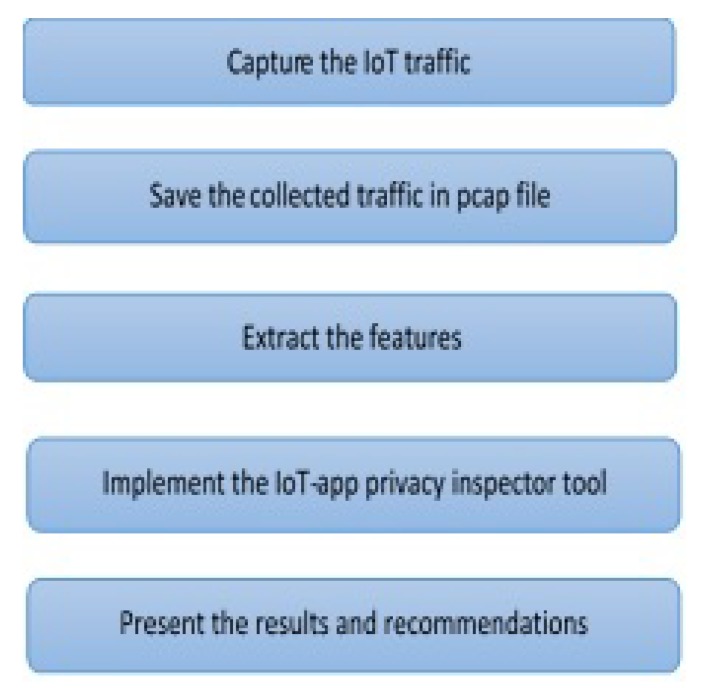
An overview of IoT-app privacy inspector tool for IoT-app user interaction type identification; identification of sensitive packet, and content type of sensitive packet identification.

**Table 1 sensors-19-04777-t001:** Type of IoT devices used in our experiments.

Type of Device	Model Type	IoT Device Manufacturer	Type of IoT-App (iOS, Android)
Smart plug	HS110	TP-Link	KASA version 2.11.0
Smart camera	NC200	TP-Link	TpCamera version 3.1.12
NetCam HD	F7D7601fc	Belkin	NetCam version 2.0.4
Smart lamp	B22	Lifx	LIFX version 3.13.0

**Table 2 sensors-19-04777-t002:** User login interaction with KASA app that controls TP-link smart plug. Methods are always invoked by the app in the order shown—top to bottom (“retrivelocation” is misspelled such as this in the packet contents). The sizes are of decrypted packets.

	Domain Name	Methods	Request Packets Size in Bytes	Response Packet Size in Bytes
Login Action	eu-wap.tplinkcloud.com	login	548	318
	api.tplinkra.com	auth token	315	278
	eu-wap.tplinkcloud.com	postPushInfo	692	178
	api.tplinkra.com	helloIotCloud	1031	435
	api.tplinkra.com	listRules	700	566
	eu-wap.tplinkcloud.com	getDeviceList	415	1143
	api.tplinkra.com	listScenes	768	568
	api.tplinkra.com	isLinked	662	817
	eu-wap.tplinkcloud.com	passthrough	520	873
	api.tplinkra.com	retriveLocation	662	574

**Table 3 sensors-19-04777-t003:** User logout interaction with KASA app that controls TP-link smart plug. Methods are always invoked by the app in the order shown—top to bottom. The sizes are of decrypted packets.

	Domain Name	Methods	Request Packets Size in Bytes	Response Packet Size in Bytes
Logout Action	eu-wap.tplinkcloud.com	logout	521	178
	api.tplinkra.com	helloIotCloud	888	427
	api.tplinkra.com	isLinked	542	772
	api.tplinkra.com	retriveLocation	542	380
	eu-wap.tplinkcloud.com	helloCloud	546	204

**Table 4 sensors-19-04777-t004:** The results of all selected classifiers based on the most common measurement; precision, recall, and F-mean.

	Packet Sensitivity Type Classifier				Packet Content Type Classifier				Interaction Type Classifier			
Classifier	P	R	F	Time	P	R	F	Time	P	R	F	Time
Decision Tree	97.1	97.1	97.1	0.093	97.1	97.1	97.1	0.088	97.1	97.1	97.1	0.072
Naive Bayes	74.0	42.1	38.8	0.043	65.3	42.05	37.6	0.035	61	51.1	49.1	0.041
K Nearest Neighbor	98.5	98.5	98.5	0.161	98.5	98.5	98.5	0.159	97.7	97.7	97.7	0.189
Multi-Layer Perception	54.2	73.6	62.4	0.873	1	71.4	83.3	1.206	52.7	72.6	84.1	1.501
Support Vector Machine	96.1	95.4	95.6	125.165	95.7	94.8	95	179.739	93.5	92.9	93.5	166.316
Random Forest	99.8	99.8	99.8	0.35	99.8	99.8	99.8	0.35	99.4	99.4	99.4	0.35

**Table 5 sensors-19-04777-t005:** Confusion matrix of the first classifier which is responsible to infer the user interaction. Rows show the actual class of a repetition and columns show the classifier’s prediction.

		Predicted Labels				
		Delete	Login	Logout	Modify Password	No-Action
True Labels	Delete	281	0	2	0	1
	Login	0	655	7	0	2
	Logout	0	0	207	0	6
	Modify Password	0	0	1	233	2
	No-action	9	0	3	0	3694

**Table 6 sensors-19-04777-t006:** Confusion matrix of the second classifier which is responsible to infer the sensitivity level of the packet. Rows show the actual class of a repetition and columns show the classifier’s prediction.

		Predicted Labels		
		Non	Non-Sensitive	Sensitive
True Labels	Non	3693	6	4
	Non-sensitive	5	699	0
	Sensitive	0	0	696

**Table 7 sensors-19-04777-t007:** Confusion matrix of the third classifier which is responsible to infer the type of the sensitive packet. Rows show the actual class of a repetition and columns show the classifier’s prediction.

		Predicted Labels				
		Non	Credential	Location	Location + Credential	User Name
True Labels	Non	3643	1	0	0	6
	Credential	0	457	0	0	1
	Location	1	0	126	0	0
	Location + Credential	0	0	0	92	0
	User name	6	0	1	0	769

**Table 8 sensors-19-04777-t008:** The accuracy of the training data and the testing data among the three classifiers.

	Packet Sensitivity Type Classifier	Packet Content Type Classifier	Interaction Type Classifier
Train accuracy	99.9%	99.9%	99.7%
Test accuracy	99.8%	99.8%	99.4%

**Table 9 sensors-19-04777-t009:** Summary of the IoT-app privacy inspector tool results on the IoT apps interactions.

	Sensitivity Level of the Packet			Content Type of the Sensitive Packet		
	User Interaction	Sensitive PII	Non-Sensitive PII	User Credentials	User Location	Username or Email Address
KASA app	Login	✓	×	✓	✓	×
	Logout	×	✓	×	×	✓
	Delete	×	✓	×	×	✓
	Change Password	✓	✓	✓	×	✓
TpCam app	Login	✓	×	✓	✓	×
	Logout	×	✓	×	×	✓
	Delete	×	✓	×	✓	✓
	Change Password	✓	✓	✓	×	✓
Netcam app	Login	✓	×	✓	×	×
	Logout	×	×	×	×	×
	Delete	×	✓	×	×	✓
	Change Password	✓	×	✓	×	×
Lifx app	Login	✓	✓	✓	×	✓
	Logout	×	✓	×	×	✓
	Delete	×	✓	×	×	✓
	Change Password	✓	✓	✓	×	✓

**Table 10 sensors-19-04777-t010:** Comparison between the IoT apps user interactions to find out which IoT apps send excessive sensitive PII about their user.

IoT Apps	User Interactions	Sensitive PII	Non-Sensitive PII
KASA app	Login	✓	×
	Logout	×	✓
	Delete	×	✓
	Change Password	✓	✓
TpCam app	Login	✓	×
	Logout	×	✓
	Delete	×	✓
	Change Password	✓	✓
NetCam app	Login	✓	×
	Logout	×	×
	Delete	×	✓
	Change Password	✓	×
Lifx app	Login	✓	✓
	Logout	×	✓
	Delete	×	✓
	Change Password	✓	✓

**Table 11 sensors-19-04777-t011:** The Accuracy results of IoT-app privacy inspector of inferring user interaction, packet level of sensitivity, and packet content type.

IoT-App Privacy Inspector		Accuracy	F1 Score
User Interaction Classifier	Login	99.4	0.994
	Logout		
	Delete		
	Change Password		
Packet Level of Sensitivity Classifier	Sensitive PII	99.8	0.998
	Non-Sensitive PII		
Packet Content Type Classifier	User Credential	99.8	0.998
	User Location		
	User name or Password		

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
