# Peer review of "Detecting IoT User Behavior and Sensitive Information in Encrypted IoT-App Traffic"

_sensors, 2019, doi:10.3390/s19214777_

Round 1

Reviewer 1 Report

This is an interesting paper. The authors propose a tool called IoT-app privacy inspector that can automatically infer the following from the IoT network traffic: the type of app-device interaction, sensitive personal information and the type of information that the traffic contains. They develop the privacy inspector tool with three multi-class classiffication methods, with each one used to infer a different goal. The tool is evaluated by using some specifically collected dataset.

Strength:
1) Motivation is good and problem is well articulated, and Fig.1 is very nice.

2) The rationales underlying the choices of the components in the tool are given.

3) The experiments results seem to be reasonable and practical.

Weakness:

1) In the experiments part, why not use some counterpart methods (e.g., [38]) as a comparison? Although the research [38] focuses on the traffic that goes directly from the IoT device to the IoT cloud (D-C) Path A in Figure 1, whereas the current work focuses on the traffic that goes directly from the IoT app that control the IoT device to the IoT cloud (A-C) path C in Figure 1. These two methods can somehow be compared.

2) The evidence for password reuse is not sufficient. The authors quote a survey (from a URL) that reveals that "52% of users use the same passwords for different services". Actually, rigorous password research show that, the percentage of users that use the same passwords for different services is about 34.02%~71.11%.

"Targeted Online Password Guessing: An Underestimated Threat." Proc. of ACM CCS 2016, pp. 1242
–1254.
"The tangled web of password reuse", In Proc. NDSS 2014.
"fuzzyPSM: A new password strength meter using fuzzy probabilistic context-free grammars" In Proc. IEEE/IFIP DSN 2016,

3) The language can be improved. Some paragraphs are too long. There are a number of typos, "curry"--->carry, "chose"--->choose, etc.

Considering the merits of the paper, I suggest a “major revision'' to this paper.

Author Response

Thank you for your valuable feedback. Please see the following response to each point individually. 

Reviewer Comment 1: In the experiments part, why not use some counterpart methods (e.g., [38]) as a comparison? Although the research [38] focuses on the traffic that goes directly from the IoT device to the IoT cloud (D-C) Path A in Figure 1, whereas the current work focuses on the traffic that goes directly from the IoT app that controls the IoT device to the IoT cloud (A-C) path C in Figure 1. These two methods can somehow be compared.

Response:

The main difference to [38] is that they only analyze the traffic rate, whereas we analyze more detailed information about the packets (size and sequence of packets). This enables us to infer more information about the user.
We have now included this discussion In Section 2.2 (IoT Privacy Concerns) in the literature review. We have modified the paragraph that starts with “Our work is similar to them …”, just below the list, to the following:

“Our work is similar to the above in that we also study and analyze the encrypted traffic of the IoT devices. However, in that research, the focus was on 1) studying the traffic that goes directly from the IoT device to the IoT cloud (D-C), i.e. Path A in Figure 1, 2) studying only the traffic rate pattern to infer the type and the activity of the IoT device. In contrast, our focus is on the traffic that goes directly from the IoT app that controls the IoT device to the IoT cloud (A-C), i.e. path C in Figure 1. Also, we do a more in-depth analysis by examining the size and sequence of the packets, and because of this we are able to infer the user interaction with the IoT device (e.g. login to the IoT device), the user sensitive data, and finally the type of such sensitive data (e.g. password). “

Reviewer Comment 2: The evidence for password reuse is not sufficient. The authors quote a survey (from a URL) that reveals that "52% of users use the same passwords for different services". Actually, rigorous password research show that, the percentage of users that use the same passwords for different services is about 34.02%~71.11%.

"Targeted Online Password Guessing: An Underestimated Threat." Proc. of ACM CCS 2016, pp. 1242 – 1254. 
"The tangled web of password reuse", In Proc. NDSS 2014.
"fuzzyPSM: A new password strength meter using fuzzy probabilistic context-free grammars" In Proc. IEEE/IFIP DSN 2016.

Response:

We have now modified the paragraph "According to different surveys conducted by [23,24,25], 53% of Internet users... " and we have added the following (Introduction, below the bullet list):
“According to Wang et al.[23, 24], 77.38% of users reuse one of their existing passwords. Also, Das et al. [25] estimate that 56% of users change their password at least once every 6 months because they tend to have the same passwords.”

Reviewer Comment 3: The language can be improved. Some paragraphs are too long. There are a number of typos, "curry"--->carry, "chose"--->choose, etc.

Response:

We have fixed these and all other typos we could find. We have shortened a number of paragraphs.

Reviewer 2 Report

I am in general happy with the research methodology and the writing.

There are some typos, grammar mistakes such as "curry", "the packet that reveal(s)", "emphasis (sizes)", "observer who access(es)", "cho(o)se", "Using accuracy to measure the performance of a classifier consider a problem", etc.

I do not think that "We believe..." is a scientific term.

I think it is not necessary to repeat the three categories so many times. 

It would be better to more clearly list the differences between the D-C and A-C models so that your contributions are clearer.

The authors claim that their tool will help the IoT user prevent any privacy violation, I do not agree because they can only detect it but the manufacturers/developers should prevent the leakage. For instance, the authors can refer to the literature on physical unclonable functions (PUFs), see Gassend or Suh and Devadas or Gunlu et al. on PUFs, as a possible solution to the leakage.

Author Response

Thank you for your valuable feedback. Please see the following response to each point individually. 

Reviewer Comment 1: There are some typos, grammar mistakes such as "curry", "the packet that reveal(s)", "emphasis (sizes)", "observer who access(es)", "cho(o)se", "Using accuracy to measure the performance of a classifier consider a problem", etc.

Response: 

We have fixed these and all other typos we could find.

Reviewer Comment 2: I do not think that "We believe..." is a scientific term.

Response:

We have deleted the words "We believe..." from the text.

Reviewer Comment 3: I think it is not necessary to repeat the three categories so many times.

Response:

To reduce repetition, we have now deleted the last sentence in the Literature review and replaced it with “to infer the three different classifications mention earlier.” If the reviewer would like to suggest other places where the repetition does not help, we are happy to make more modifications. We originally repeated the three categories throughout the text so the reader would not need to remember what they are.

Reviewer Comment 4: It would be better to more clearly list the differences between the D-C and A-C models so that your contributions are clearer.

Response: In order to clearly emphasize on our contribution, in Section 2.2. (IoT Privacy Concerns) in the literature review, we have now modified the paragraph that starts with “Our work is similar to them …” just below the list, to the following:

“Our work is similar to the above in that we also study and analyze the encrypted traffic of the IoT devices. However, in that research, the focus was on 1) studying the traffic that goes directly from the IoT device to the IoT cloud (D-C), i.e. Path A in Figure 1, 2) studying only the traffic rate pattern to infer the type and the activity of the IoT device. In contrast, our focus is on the traffic that goes directly from the IoT app that controls the IoT device to the IoT cloud (A-C), i.e. path C in Figure 1. Also, we do a more in-depth analysis by examining the size and sequence of the packets, and because of this we are able to infer the user interaction with the IoT device (e.g. login to the IoT device), the user sensitive data, and finally the type of such sensitive data (e.g. password).“ 

Reviewer Comment 5: The authors claim that their tool will help the IoT user prevent any privacy violation, I do not agree because they can only detect it but the manufacturers/developers should prevent the leakage. For instance, the authors can refer to the literature on physical unclonable functions (PUFs), see Gassend or Suh and Devadas or Gunlu et al. on PUFs, as a possible solution to the leakage.

Response:

Throughout the text, we have made appropriate modifications to not give the impression that the tool prevents privacy violations.

We read the literature of PUF, but we did not understand how this is relevant to our research. 

Reviewer 3 Report

The article is correctly constructed methodologically with division into thematic sections. The problem of Device -to-cloud and App-to -cloud traffic generation has been analyzed for sensitive data leaks. The authors proposed three IoT network traffic classifiers. The conducted experiments proved the thesis about the possibility of correct understanding and individual types of message patterns. A well and completely developed literature review containing the most important items about IoT Traffic Classification.

Author Response

Thank you for your valuable feedback.

There are no comments from the reviewer to response to. 

Round 2

Reviewer 1 Report

This is a nice paper. Thanks for the good replies. I still have one question: The proposed IoT-app privacy inspector can be also misused by attackers to breach user privacy. The inspector needs to perform a Man in the Middle attack (MITM) between the smartphone app and the cloud. I am wondering how to prevent such IoT-app privacy inspector from been misused? Can the following third-party mutual authentication schemes prevent such kind of MITM attack (and thus prevent the IoT-app privacy inspector)? "Efficient and Anonymous Mobile User Authentication Protocol Using Self-certified PKC for Multi-server Architectures", IEEE TIFS, 2016. "Measuring Two-Factor Authentication Schemes for Real-Time Data Access in Industrial Wireless Sensor Networks".IEEE Transactions on Industrial Informatics, 2018. I suggest an "accept with minor revision".

Author Response

Thank you for your feedback, please see the following response to the comment

Reviewer Comment: The proposed IoT-app privacy inspector can be also misused by attackers to breach user privacy. The inspector needs to perform a Man in the Middle attack (MITM) between the smartphone app and the cloud. I am wondering how to prevent such IoT-app privacy inspector from been misused?

We split our response to the above comment into two parts as following

comment part 1: “The proposed IoT-app privacy inspector can be also misused by attackers to breach user privacy.”

Response:

First of all, we only need to do a MITM attack to collect training data for the IoT Inspector tool, which means we only need to do MITM against devices that we own/control, so that’s not a breach of user's privacy.

Second of all, after the tool/classifier has been trained (with the attacker’s own data, which he can collect with a MITM against his own IoT devices), the only thing the attacker needs to do is to collect encrypted data from the user’s home network, hence the attacker does not need to do a MITM attack.

comment part 2: "The inspector needs to perform a Man in the Middle attack (MITM) between the smartphone app and the cloud. I am wondering how to prevent such IoT-app privacy inspector from been misused?"

Response:

The authors perform a MITM attack between the smartphone app and the IoT cloud to collect the encrypted traffic for in-depth analysis and to train the machine learning. Hence, the IoT-app privacy inspector does not perform MITM by itself. Instead, it accepts a collection of encrypted traffic from different IoT devices in a PCAP format. Then it applies supervised machine learning to classify the encrypted traffic on three different categories (based on the size and sequence).